# Diagnostic Testing Accuracy for *Helicobacter pylori* Infection among Adult Patients with Dyspepsia in Cuba’s Primary Care Setting

**DOI:** 10.3390/microorganisms11040997

**Published:** 2023-04-11

**Authors:** Amílcar Duquesne, Rosabel Falcón, Belkys Galindo, Onelkis Feliciano, Oderay Gutiérrez, Waldemar Baldoquín, Magile C. Fonseca, Rafael Llanes, Luis Sarmiento

**Affiliations:** 1Teaching Orthopedical Hospital Fructuoso Rodríguez, Havana City 10400, Cuba; 2Department of Microbiology, Center for Research, Diagnosis and Reference, “Pedro Kourí” Institute of Tropical Medicine, Havana City 11400, Cuba; 3Department of Epidemiology, Center for Research, Diagnosis and Reference, “Pedro Kourí” Institute of Tropical Medicine, Havana City 11400, Cuba; 4Department of Virology, Center for Research, Diagnosis and Reference, “Pedro Kourí” Institute of Tropical Medicine, Havana City 11400, Cuba; 5Department of Clinical Sciences, Immunovirology Unit, Skåne University Hospital, Lund University, 21428 Malmö, Sweden

**Keywords:** *Helicobacter pylori*, diagnostic accuracy, primary care

## Abstract

Evidence of the effectiveness of the tests used to diagnose *Helicobacter pylori* (*H. pylori*) in primary healthcare is limited. This cross-sectional study aims to assess the accuracy of tests used for to diagnose *H. pylori* infection in primary care patients and its relationship with gastroduodenal pathologies. Over 12 months, 173 primary care patients with dyspeptic symptoms were referred for upper gastrointestinal endoscopy to obtain gastric biopsies, and venous blood was extracted from them. *H. pylori* infection was detected using a rapid urease test (RUT), real-time polymerase chain reaction (RT-PCR), *H. pylori*-IgG ELISA, and Western blot (WB). The culture and histological findings were used as the reference standard for *H. pylori* infection. *H. pylori* prevalence was 50%. There were no significant differences between men and women overall or by age group. The presence of *H. pylori* was associated with chronic moderate gastritis and its absence with chronic inactive gastritis, as well as the combination of gastritis and gastric lesions (*p <* 0.05). RUT and ELISA *H. pylori* -IgG tests showed the highest overall performance (accuracy 98.9% and 84.4%), followed by WB and RT-PCR (accuracy 79.3% and 73.9%). These findings support the notion that combined invasive and noninvasive methods, such as RUT and *H. pylori*-IgG ELISA, can be a primary diagnostic screening tool for detecting *H. pylori* among adult dyspeptic patients in Cuba’s primary care setting.

## 1. Introduction

*Helicobacter pylori* (*H. pylori*) is a common bacterium, infecting 50% of the human population worldwide and almost 90% in developing countries. It is considered one of the most prevalent chronic bacterial infections in humans [1], associated with gastritis, duodenal ulcer, gastric carcinoma, and mucosa-associated lymphoid tissue [2]. In 2014, the International Agency for Research on Cancer classified *H. pylori* infection as a class 1 carcinogen. In 2017, the World Health Organization raised an alert about *H. pylori* resistance to clarithromycin, the most crucial antibiotic in combined therapy for eradicating this microorganism [3,4].

*H. pylori* infection can be diagnosed by several invasive (e.g., histology, rapid urease test [RUT], bacterial culture from biopsy) and noninvasive (e.g., polymerase chain reaction, serological tests) techniques [3]. Among noninvasive techniques, the guidelines for detecting and managing *H. pylori* consider the urea breath test (UBT) and the stool antigen test (SAT) as valid options for initial screening in dyspeptic patients [5,6,7]. The choice of diagnostic test depends on the prevalence of *H. pylori*, the local incidence of age-related gastric cancer, the advantages and disadvantages of each method, the different clinical conditions for each patient, and the test costs [8].

Multiple factors, such as socioeconomic level, age, hygiene, and sanitary habits, can influence *H. pylori* infection rates in developing countries [1,6]. Some studies [9,10,11,12] have suggested that the mean prevalence of *H. pylori* infection in Cuban adults is similar to that reported in the Latin America and Caribbean region (59.3 %, 95% CI 52.9–65.6) [1]. About 1300 deaths per year in Cuba are attributed to peptic ulcer (*n* = 366) and stomach cancer (*n* = 865), and the overall death rate is higher for men older than 40 years of age [13]. Therefore, public health policies have been directed toward the early detection of gastroduodenal diseases associated with *H. pylori* by endoscopic and histopathological methods [9]. Although the biopsy-based direct methods have been accepted as reference standards for *H. pylori* infection diagnosis, these tests are considered inappropriate for routine screening. The requirement of mandatory biopsy implies more discomfort for the patient and demands skillfulness in the performance and interpretation [3,5]. In addition, the evaluation of other diagnostic tests remains poorly explored at the primary care level. 

Because of all these reasons, strategies for community-based diagnoses should be established based on local epidemiology and the availability of diagnostic tools. Hence, this study assesses the accuracy of tests in diagnosing *H. pylori* infection, in agreement with the evidence-based diagnostic criteria [14] and its relationship with gastroduodenal pathologies. The aim is to determine which diagnostic tests are more accurate when applied to adult patients with dyspepsia in Cuba’s primary care setting.

## 2. Materials and Methods

### 2.1. Ethical Considerations

The ethical committee of the “Pedro Kourí” Institute of Tropical Medicine (IPK) approved this study (approval number 2014/08). The study was performed in compliance with ethical principles outlined in the Declaration of Helsinki and was consistent with Good Clinical Practice guidelines. All patients provided informed written consent for inclusion before they participated in the study, and the data obtained from each patient was kept confidential. The information in this study was used only for research purposes.

### 2.2. Study Area, Patients and Design

This is a cross-sectional diagnostic cohort study [15] aiming to evaluate the accuracy of invasive and noninvasive tests in samples from Cuban adults with dyspeptic symptoms. The study was carried out between November 2016 and November 2017 in the Health Area of Primary Care Polyclinic 19 de Abril, Plaza de la Revolución Municipality, Havana, Cuba. Health and academic institutions such as IPK, Teaching Surgical Clinical Hospital Manuel Fajardo, and Charité University also participated in this study.

One hundred and seventy-three patients of both sexes, aged >18 years, with uninvestigated dyspepsia (defined by any of the following symptoms: postprandial fullness; early satiety; epigastric pain; and epigastric burning [16]), and with the capacity to consent for themselves and interest in participating in the study were enrolled for routine upper gastrointestinal endoscopy and considered potentially eligible. Exclusion criteria comprised those who have received previous treatments with antimicrobials, proton pump inhibitors, non-steroidal anti-inflammatory drugs, and bismuth salts, as well as those with a history of digestive bleeding within three–four weeks before enrollment. Patients to whom a biological sample could not be obtained or with poor-quality samples were also excluded. All eligible’ patients were exposed to the same index tests and reference standards at the same time point.

The study is in agreement with the “Strengthening the Reporting of Observational Studies in Epidemiology” (STROBE) statement guidelines [17]. The STROBE checklist is included in Appendix A.

### 2.3. Specimen Collection and Samples Treatment

Four biopsy samples of each patient were taken from the lesions and areas around the pyloric antrum and gastric body by experienced gastroenterologists using an Olympus fibroendoscope [9]. The examination was conducted under topical anesthesia following at least 8 h of fasting. The first biopsy sample was taken for histopathology; the second biopsy sample was taken for culture. Two additional biopsies samples were taken for RUT and real-time polymerase chain reaction (RT-PCR). In addition, 10 mL of venous blood was collected from the patients, processed according to the technical standards established for obtaining serum, and stored at −20 °C until its use for serological tests. Sample processing was performed by pathologists and microbiologists with >8 years of experience to avoid variability between observers. All patient information and the test outcome were withheld until after the completion of the study. 

### 2.4. Histopathology

The biopsy samples were fixed in 10% formalin solution, cut into slides, and stained with hematoxylin and eosin (H&E) and giemsa. The Sydney and OLGA/OLGIM classifications systems were used for histological characterization of the pathologies: chronic gastritis, peptic ulcer, and gastric lesions (intestinal metaplasia, gastric dysplasia, atrophic gastritis, and gastric adenocarcinoma) [18]. This method was performed independently on all patients to ensure the reliability of the estimates. Two pathologists examined each glass slide blindly for the presence/absence of *H. pylori*. The results were corroborated by a third pathologist to control for reporting bias. 

### 2.5. Culture 

One biopsy sample was grown on Columbia agar plates with Columbia blood agar medium (Oxoid, Hampshire, UK), containing 10% sheep-defibrinated blood and 1% heat-inactivated fetal bovine serum, supplemented with Dent antibiotic supplement (Oxoid, Hampshire, UK). The plates were incubated at 37 °C under microaerophilic conditions (10% CO_2_, 5% O_2,_ and 85% N_2_) (CampyGen Compact, Oxoid, Hamphire, UK) with saturated humidity for five days. Morphological identification by Gram staining (Gram-negative bacilli), and biochemical oxidase, catalase, and urease tests were performed as described by Llanes et al. [9].

### 2.6. RUT 

A biopsy specimen from each patient was inoculated into a vial containing 0.3 mL of urease test broth (Stuart’s transport medium, BBL, Cockeysville, MD, USA) at room temperature. A positive test was considered by the color change from the original yellowish to fuchsia [19].

### 2.7. H. pylori-IgG ELISA 

The detection of IgG-class antibodies to *H. pylori* was performed using a commercial ELISA test kit (IBL International, Hamburg, Germany), previously validated at the IPK [20]. 100 μL of negative control, positive control, and serum samples were prepared according to the manufacturer’s instructions and added to each well of the ELISA microplate. The optical density was determined using a microplate reader at 450 nm. Cutoff index values of >1.2 were considered positive, values <0.8 were considered negative, and values from 0.8 to 1.2 were considered indeterminate.

### 2.8. Western Blotting (WB) 

WB qualitative assay validated at the IPK [21] was performed using the Helicoblot 2.1 kit (Genelabs Diagnostics, Singapore) following the manufacturer’s instructions [22]. Immunoreactive bands were scanned using a GS-800 Calibrated Densitometer; (BioRad Laboratories, Hercules, CA, USA) equipped with Quantity One software (version 4.6.2, Bio-Rad). The presence of *H. pylori* recombinant proteins (89 kDa [VacA], 116 kDa [CagA], 37 kDa, 35 kDa, 30 kDa, 19.5kDa) with or without the current infection marker (10 kDa) was considered a positive result.

### 2.9. RT-PCR 

DNA from biopsy specimens was extracted using the Macherey-Nagel kit (GmbH & Co. KG, Germany). A 267 base-pair fragment of the 23S ribosomal RNA gene was amplified using a LightCycler 480 thermal cycler version LCS480 1.5.1.62 (Roche Diagnostics, France) [23]. Each run used 10^−2^–10^−6^ dilutions of DNA (45-mg/mL) of the Hp H37Rv reference strain as a positive control and sterile water as a negative control. RT-PCR program consisted of one cycle of 10 min at 95 °C, followed by 50 cycles of 10 s at 95 °C, 10 s at 60 °C, and a final cycle of 17 s at 72 °C. Cycle number threshold values between 17–33 were considered positive when the negative control was undetectable. 

### 2.10. Definition of H. pylori Status 

A diagnostic positive was defined based on culture and histopathology positives. If at least one of the tests was negative, the *H. pylori* infection status was considered negative. 

### 2.11. Statistical Analysis 

The sample size was estimated through paired design analysis based on the McNemar test using Epidat Software version 3.1 [24]. Sample recruitment was guided by an expected 59% prevalence of *H. pylori* infection in a screening cohort, based on previous national and regional published data [11,12,21,25]. Calculations were performed using an expected sensitivity of 80% to detect sensitivity differences between the pairs (18%). A minimum sample size of 84 participants was deemed sufficient for 80% power and a significance level of α = 0.05. 

Personal data (age, sex), diagnostic test results, *H. pylori* status definition (Appendix A), and endoscopic and histopathological reports were included in the patient’ inquiry form. Absolute and relative frequencies were determined for each variable studied. The association between *H. pylori* infection and each diagnosis was studied using Pearson’s chi-squared test of independence and a significance level α = 0.05. Sensitivity, specificity, positive and negative predictive values, positive and negative likelihood ratios, diagnostic accuracy with 95% confidence intervals (CIs), and the Youden index of each test were calculated using the Epidat program (Appendix A). Kappa values over 0.61 were considered a good agreement. 

## 3. Results

After excluding 81 patients who did not meet the criteria for eligibility, 92 patients were included in the diagnostic test analysis. They were grouped according to the *H. pylori* status definition (infected patients, *n* = 46; non-infected patients, *n* = 46) (Figure 1).

Patients included in the study were, on average, 50 years old. The most frequent age range was 41–50 years (26.1%), followed by 60–87 (23.9%), and 51–60 (22.8%). Sixty-two percent of the participants (57) were females and 38% (35) were male, with a female/male ratio of 1.62:1.00 (Table 1). 

According to the status definition, *H. pylori* infection prevalence was 50%, with a female predominance of 58.7%. The mean age of the patients who tested positive for *H. pylori* was 51.08 ± 15.49, with predominance in the age groups over 41 years (82.6%). There were no significant differences between men and women overall (*p* = 0.6676), or by age group (*p* = 0.1343) (Table 1).

Chronic gastritis (31.5%) and combined gastritis and duodenitis (29.3%) were the most frequent endoscopic and histopathological findings. The presence of *H. pylori* (12/26.1%) was associated with chronic moderate gastritis (*p* = 0.0007) and its absence was associated with chronic inactive gastritis (*p* = 0.0102) and the combined classification of gastritis and gastric lesions (*p* = 0.0352) (Table 2).

RUT showed the highest overall performance with a sensitivity of 98.9%, specificity of 98.9%, accuracy of 98.9%, Youden index of 1.0, and kappa coefficient of 0.98, followed by Hp-IgG ELISA and WB. The RT-PCR showed the lowest performance with a sensitivity of 76.1%, specificity of 71.7%, total accuracy of 73.9%, Youden index of 0.5, and kappa coefficient of 0.48. RUT revealed the lowest false negative and false positive results and the best probability of correctly predicting the disease’s presence and/or absence (Table 3).

## 4. Discussion

*H. pylori* infection can be diagnosed by invasive and noninvasive techniques, such as RUT, culture, histopathology, RT-PCR, UBT, SAT, and serology. Each test has benefits and limitations; none is considered a perfect reference standard whose choice is decisive in assessing a diagnostic test’s accuracy [3,5,7]. Access to endoscopy for upper gastrointestinal tract diseases in Cuba is public at the primary healthcare level; however, some diagnostic tests are restricted to specialized hospitals and institutions. Moreover, the strategies for an effective diagnostic approach, particularly in primary care, depend on the clinical situation, local epidemiology, and cost effectiveness of diagnostic tests [8,26,27]. 

In the present study, we accurately evaluated noninvasive and invasive tests using samples from Cuban adult patients with dyspeptic symptoms. Consequently, RUT and *H. pylori*-IgG ELISA demonstrated the highest diagnostic performance for this infection. This information is particularly relevant for primary care services because the results support using these tools to monitor *H. pylori*-related diseases in susceptible populations. 

The prevalence of *H. pylori* infection found in adults (50%) was superior to the current prevalence of *H. pylori* infection worldwide (44.3%) and similar to the *H. pylori* infection rate revealed in developing countries (50.8%) [1]. It is worth noting that a study by Galbán et al. showed an infection rate slightly higher (58.4%) in primary care facilities [10], comparable with the prevalence (59.3%) in Latin America and the Caribbean region [1]. 

This study shows a female predominance in *H. pylori* infection compared to males as, in Cuba, they are more frequent visitors to endoscopy services than males. Despite the previous statement, Agah et al. [28] found an association between the female gender and this bacterial infection. However, some investigations reveal a discrepancy in susceptibility to *H. pylori* infection on behalf of male predominance [29,30]. 

Several investigations based on geographically defined Cuban populations indicated dissimilar sociodemographic characteristics and unhealthy lifestyles [9,20,21]. This fact could explain the increase in the greater number of patients with gastritis (29/92), the gastric disease most frequently encountered in primary care in Havana [10], and the most frequent cause of chronic gastroduodenal disease in Cuba [9,21]. Likewise, other observations also showed combined gastritis and peptic ulcer in *H. pylori*-positive cases. Previous studies have demonstrated the presence of the bacterium in 95–100% of duodenal ulcers and 85–95% of gastric ulcers, suggesting an improvement in these patients when they adopt lifestyle changes and use validated anti-*H. pylori* therapies [3,6,9,31].

The *H. pylori*-negative association was corroborated in patients with combined gastritis and gastric lesions because, in these cases, the bacterial load is lowest, which negatively influences *H. pylori* detection [31,32]. However, it is essential to have tests that allow an early diagnosis of gastric lesions due to their evolution into stomach cancer, whose incidence rate in Cuba in 2020 was 9.7 for men and 6.1 for women per 100,000 inhabitants [11].

A limitation of this study was the insufficient number of biopsies analyzed in the histopathological examination. Sydney’s protocol and international consensus on *H. pylori* diagnosis support taking five gastric biopsy punches in different locations to reduce sampling errors and increase the efficacy of invasive methods [5,31,32]. However, a single biopsy sampling from the lesser curvature near the incisura angularis or the greater curvature opposite the incisura angularis has been considered for *H. pylori* infection diagnosis [33]. In this investigation, the use of specific stains (H&E and giemsa) for *H. pylori* detection, the pathologist’s expertise, and concordance with a culture made it possible to establish the current *H. pylori* status definition. Combining two available invasive tests (culture and histopathology) produced a better indicator of disease status.

The results of the studies by Galbán et al. provided evidence that RUT might be useful for diagnosing *H. pylori* infection in adult patients with gastroduodenal symptoms [10]. The present study further extends these findings and highly recommends RUT as a valuable tool to use in the endoscopy department of all Cuban primary care services due to its simple execution and straightforward interpretation compared with other invasive tests [34,35]^.^

Another test that showed good accuracy (>80%) compared to the *H. pylori* status definition was *H. pylori*-IgG ELISA. A prior evaluation through case-control analysis (109 adult dyspeptic patients and 277 healthy individuals) reported excellent performance of ELISA-IBL (91.2% sensitivity, 90.2% specificity, 93.9% positive predictive value, and 86% negative predictive value), recommending this technique to evaluate *H. pylori* infection prevalence in the population [20]. The anti *H. pylori*-IgG antibody levels correlate with high diagnostic precision [36,37], which could guide the physician when endoscopy is unavailable or unjustified. This test is considered suitable for initial screening of *H. pylori* infection in a population with a rate of infection greater than 30% and the absence of other noninvasive tests [32,38]. Hence, the IgG detection ELISA system could be a screening tool for diagnosing *H. pylori* infection at the community level. 

The commercial WB assay Helicoblot 2.1 is an excellent test for identifying different *H. pylori* antigenic markers and providing valuable prognostic information [21,22,39]. However, this assay was used as a second-line serological method to evaluate *H. pylori* seropositivity because of the potential for a false positive result. Even though WB may be a valuable tool for the direct visualization of highly specific *H. pylori* antigens, the findings of this study do not sustain the use of the Helicoblot 2.1 system as an initial screening tool in primary care centers. 

In the literature, there is a tendency towards better sensitivity for RT-PCR, essentially from gastric biopsy, as it has several advantages, such as short working time, direct detection of microorganism presence, and low risk of contamination [23,40]. However, RT-PCR used in the study is not valuable in routine practice because of the low accuracy. Ramírez et al. recommend the detection of two *H. pylori* genes for better sensitivity in diagnoses from biopsies [41]. Other studies support the use of two biopsies obtained from antrum and corpus to increase the accuracy in detecting *H. pylori* infection by RT-PCR [42,43]. 

Based on these findings, we recommend using antibody testing (*H. pylori*-IgG ELISA) in any adult with dyspepsia <40 years old without any previous diagnosis of *H. pylori* infection, use of proton pump inhibitors and antimicrobials, or elevated risk of gastric cancer. RUT could be considered together with histology as the best approach for *H. pylori* diagnosis in adults >40 years old with an indication for upper endoscopy and a family history of gastric cancer in a first-degree relative. 

In the Cuban context, the assessment of diagnosis accuracy carried out with different invasive and noninvasive tests offers an alternative to the recommended tests by international consensus. Using *H. pylori*-IgG ELISA and RUT at the primary care level would help focus available resources and yield future diagnostic evidence-based investigations that impact national guidelines for diagnosing and managing gastroduodenal diseases with *H. pylori* infection in adult patients. 

## Figures and Tables

**Figure 1 microorganisms-11-00997-f001:**
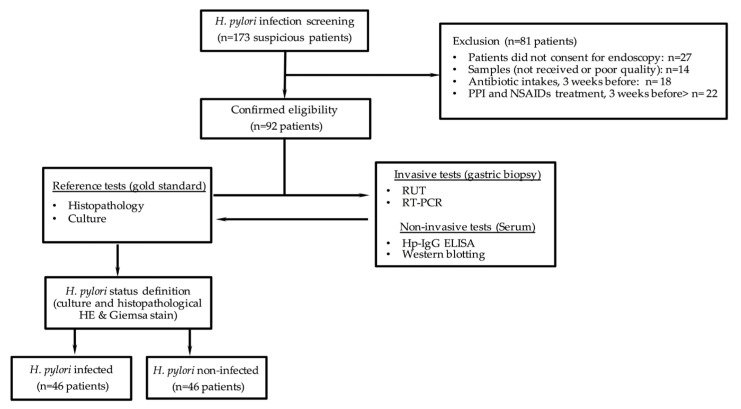
Description of eligible, enrolled, and participating patients in the cross-sectional study cohort type to assess diagnostic accuracy.

**Table 1 microorganisms-11-00997-t001:** Baseline characteristics of the participants.

Characteristic	*H. pylori*-Infected Patients ^†^ (*n* = 46)	All Patients (*n* = 92)	*p* Value
Age (median age)	39	50	
Age groups n, (%)			
≤20	-	3 (3.3)	0.1343
21–30	2 (4.3)	10 (10.9)	
31–40	6 (13.0)	12 (13.0)	
41–50	12 (26.1)	24 (26.1)	
51–60	13 (28.3)	21 (22.8)	
>60	13 (28.3)	22 (23.9)	
Sex n, (%)			
Female	27 (58.7)	57 (62.0)	0.6676
Male	19 (41.3)	35 (38.0)	

Significance level if *p* < 0.05. **^†^** Diagnosis according to the *H. pylori* status definition.

**Table 2 microorganisms-11-00997-t002:** *H. pylori* infection status according to gastroduodenal diseases diagnosed by endoscopic and histopathological examination among the study population (*n* = 92).

Histological ^†^ and Endoscopic Findings	*H. pylori* Diagnosis ^†^ n (%)		Total (*n* = 92)	χ^2^	*p* Value
	Positive (*n* = 46)	Negative (*n* = 46)			
Normal mucose	0 (0.0)	2 (4.3)	2 (2.2)	0.0000	1.0000
Chronic gastritis	18 (39.1)	11 (23.9)	29 (31.5)	1.8128	0.1782
Mild	4 (8.7)	1 (2.2)	5 (5.4)	0.8460	0.3577
Moderate	12 (26.1)	0 (0.0)	12 (13.0)	11.5958	0.0007 *
Severe	2 (4.3)	0 (0.0)	2 (2.2)	0.5111	0.4747
Inactive	0 (0.0)	10 (21.7)	10 (10.9)	6.6083	0.0102 *
Combined gastritis and duodenitis	14 (30.4)	13 (28.3)	27 (29.3)	0.0000	1.0000
Combined gastritis and peptic ulcer	11 (23.9)	4 (8.7)	15 (16.3)	2.8675	0.0904
Combined gastritis and gastric lesions	0 (0.0)	8 (17.4)	8 (8.7)	4.4337	0.0352 *
Combined gastritis and peptic ulcer and gastric lesions	1 (2.2)	2 (4.3)	3 (3.3)	0.0000	1.0000
Combined gastritis and duodenitis and peptic ulcer	2 (4.3)	0 (0.0)	2 (2.2)	0.5111	0.4747
Combined gastritis and duodenitis and gastric lesions	0 (0.0)	6 (13.0)	6 (6.5)	2.4739	0.1157

***** Significance level according to χ^2^ test with Yate’s correction (*p* < 0.05). **^†^** Gastroduodenal diseases according to histological findings based on the Sydney and OLGA/OLGIM classifications.

**Table 3 microorganisms-11-00997-t003:** Precision indicators of diagnostic tests for detecting *H. pylori* in adult Primary Healthcare patients with the gastroduodenal disease compared to *H. pylori* status definition.

Type	Test	Parameters Expressed by % (95% CI)								
		Sensibility	Specificity	PPV	NPV	PLR	NLR	Accuracy	YI	Kappa
Invasive (biopsy sample)	RUT	98.9 (90.5–99.9)	98.9 (90.3–99.9)	98.9 (90.5–99.9)	98.9 (90.3–99.9)	92.0 (5.7–1417.7)	0.01 (0.0–0.2)	98.9 (94.1–99.8)	1.00	0.98
	RT-PCR	76.1 (62.1–86.1)	71.7 (57.5–82.7)	72.9 (59.0–84.7)	75.0 (60.6–85.4)	2.7 (1.7–4.4)	0.3 (0.2–0.6)	73.9 (64.1–81.8)	0.5	0.48
Noninvasive (serum sample)	*Hp*-IgG ELISA	97.8 (88.4–99.6)	71.1 (56.6–82.3)	77.2 (64.8–86.2)	97.1 (84.7–99.5)	3.4 (2.1–5.4)	0.03 (0.0–0.2)	84.4 (75.6–90.5)	0.7	0.65
	WB	89.4 (77.4–95.4)	68.9 (54.3–80.5)	75.0 (62.3–84.5)	86.1 (71.3–93.9)	2.9 (1.8–4.5)	0.2 (0.1–0.4)	79.3 (70.0–86.4)	0.6	0.58

Rapid urease test (RUT); RT-PCR (real-time polymerase chain reaction); WB (Western blotting); confidence interval (CI); positive predictive value (PPV); negative predictive value (NPV); positive likelihood ratio (PLR); negative likelihood ratio (NLR); Youden index (YI).

## Data Availability

The data presented in this study are available from the corresponding author upon reasonable request.

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
