# Peer review of "Diagnostic Testing Accuracy for Helicobacter pylori Infection among Adult Patients with Dyspepsia in Cuba’s Primary Care Setting"

_microorganisms, 2023, doi:10.3390/microorganisms11040997_

Round 1

Reviewer 1 Report

Dear Authors 

since you performed an observational study I strongly recommend to follow the STROBE checklist and describe in the text all the included items.

You can finde the list at

https://www.strobe-statement.org/download/strobe-checklist-cross-sectional-studies-pdf

The sample size should be carefully justified and on the basis of previous study and the power of the statistical analysis should be reported as well

Author Response

We thank the reviewer for the thorough review, which helped improve our manuscript, and we hope to have addressed all issues raised appropriately.

Reviewer 2 Report

This manuscript is describing about diagnostic testing accuracy for Helicobacter pylori infection  in Cuban adult primary care patients with dyspepsia. The authors screened 173 primary care patients and confirmed elegibility with 92 patients. They found that combined invasive and noninvasive methods, such as the RUT and H. pylori-IgG ELISA, can be a primary diagnostic screening tool for detecting H. pylori in patients with dyspeptic symptoms. Though the sample size was not bog, data were interesting and the manuscript was written well and it is recommended to be acceptable.

Author Response

We thank the reviewer for this summary and comments on the limitations of our study.

Reviewer 3 Report

The specific comments are listed in detail as follows:

1. P4, lines 98, 113, 124, 138, and 142, “H. pylori” should be italicized. Why are “was done under topical” in line 86,positive in line 102, and “Cutoff index values of >1.2 were considered positive, values <0.8 were considered negative, and values from 0.8 to 1.2 were considered indeterminate” in lines 117~119 italics? The authors are advised to revise the full text carefully

2. Lines 105~106, “One biopsy sample was grown on Columbia agar plates Columbia blood agar me-105 dium (Oxoid, UK)” is advised to revise to “One biopsy sample was grown on Columbia agar plates with Columbia blood agar me-105 dium (Oxoid, UK)”.

3. Line 206, The reference in the sentence of “in primary care 206 facilities 9.” are labeled in a way that is inconsistent with other references.

4. The data provided in this article is insufficient. The authors should provide the specific test data of the four test methods including RUT, RT-PCR, Hp-IgG ELISA, and WB, and make these data into a supporting material publishing together with this paper.

5. For the readers to know more information, it is suggested that the calculation methods of the data in Table 3 such as “Confidence interval (CI), Positive Predictive Value (PPV), Negative Predictive Value (NPV), Positive Likelihood Ratio (PLR), Negative Likelihood Ratio (NLR), Youden index (YI)” should be provided in the supporting materials.

Author Response

(The authors gave the same response as above.)

Round 2

Reviewer 1 Report

Dear Authors

All the relevant points have been addressed

Reviewer 3 Report

I have evaluated the revised version of the manuscript (Microorganisms-2252284). Authors have evaluated the reviewers’ and editor’s comments and suggestions, responded to the suggestions point-by-point, and revised the manuscript accordingly. I think that this manuscript Microorganisms-2252284 is suitable for publication on Microorganisms.